# The Microstructure Evolution and Mass Transfer in Mushy Zone during High-Pressure Solidifying Hypoeutectic Al-Ni Alloy

**Xiaohong Wang** [1], **Duo Dong** [1], **Dongdong Zhu** [1,*], **Hongwei Wang** [2] and **Zunjie Wei** [2,*]

[1] Key Laboratory of Air-Driven Equipment Technology of Zhejiang Province, Quzhou University, Quzhou 324000, China; hitxiaohong_wang@hotmail.com (X.W.); dongduohit@163.com (D.D.)

[2] National Key Laboratory of Metal Precision Hot Forming, Harbin Institute of Technology, Harbin 150001, China; wanghw@hit.edu.cn

\* Correspondence: zhudd8@163.com (D.Z.); weizj@hit.edu.cn (Z.W.); Tel.: +86-570-802-6634 (D.Z.); +86-138-0461-0088 (Z.W.)

**Abstract:** The mushy zone of hypoeutectic Al-1.5 wt% Ni alloy during high-pressure synthesis was obtained by changing the structure of the graphite heater. Meanwhile, the evolution of the microstructure was investigated. The results demonstrated that three distinguished zones were successfully generated along with the direction of the temperature gradient, including the fully melted area consisting of the columnar dendrite. The mushy zone is composed of an $\alpha$-Al phase, bulk $\beta$-Al$_3$Ni phase, and a eutectic microstructure as well as a non-melted, solid region. In addition, the mass transfer velocity and the time required for the liquid pool to migrate through the mushy zone during thermal-stable treatment under different high pressures were also analyzed. The results showed that the mass transfer was greatly inhibited, and the minimum time required for a liquid droplet to go through the whole mushy zone at 1 GPa and 3 GPa was 746 h and 5523 h, respectively.

**Keywords:** Al alloy; high pressure; high temperature; mushy zone; mass transfer

## 1. Introduction

The temperature gradient is in the nature of every solidification process [1–3]. For alloys with a certain solidification range, it is inevitable to generate a mushy zone, which is characterized by partial melting during cooling down. Furthermore, solidification microstructures (i.e., the shape [4–6], size [7–9], composition [10–12]) formed in the mushy zone determine the final performance of the material to a great extent [13,14]. Accordingly, it is significant to study the characteristics of the mushy zone during the solidification process.

High-pressure solidification is featured by application of high pressure during the melting and solidification process. Numerous works have proven that application of high pressure can greatly influence the solidification parameters, for example, inhibition of solute diffusion [15] and decrease in interfacial solidification velocity [16]. The temperature distribution in a cylindrical furnace for a high-pressure experiment has also been researched [17,18], and the results demonstrated that the temperature distribution is uneven and the temperature gradient is normally relatively low at the solidification interfaces. Hence, it provides the possibility to study the microstructure evolution resulting from the uneven distribution of temperature during high-pressure solidification.

Al-Ni intermetallic compounds have excellent corrosion resistance, high temperature stability, and strong corrosion resistance, which cannot be replaced by other alloys. Therefore, Al-Ni intermetallic compounds are regarded as the research objects of national defense and the military industry. For the Al-Ni alloy phase diagram [19], there are eutectic reactions at the Al-rich and Ni-rich ends, which are

L → α-Al + Al$_3$Ni and L → α-Ni + AlNi$_3$, respectively. A series of peritectic reactions exist in the middle part. There are also abundant intermetallic compounds with different solid solubility ranges in the alloy system, such as Al-Ni, Al$_3$Ni, AlNi$_3$, Al$_3$Ni$_5$ and Al$_3$Ni$_2$. According to their solid solubility, the above intermetallic compounds can be simply divided into two categories: intermetallic compounds with large solid solubility and low melting entropy, such as Al-Ni, and intermetallic compounds with small/nil solid solubility and high melting entropy, such as Al$_3$Ni. The liquid–solid correlation, phase structure, and rapid solidification kinetics of Al-Ni alloys have been extensively studied [20–22], which provide a theoretical basis for the study of microstructure evolution and phase growth of Al-Ni alloy under high pressure.

In order to explore the mushy zone solidification behavior of Al-Ni alloy under high pressure, the structure of a graphite heater for high-pressure synthesis was redesigned to produce a temperature gradient along the radial direction of the sample, thus forming a mushy zone. In addition, the migration behavior of the liquid pools and channels of the mushy zone with the effect of high pressure was studied.

## 2. Materials and Methods

For the experiment, Al-1.5 wt% Ni alloy was prepared by conventional casting from pure Al (99.99 wt%) and Ni (99.99 wt%). The samples for high-pressure solidification were cylinders of 20 mm in diameter and 18 mm in length. A tungsten carbide six-anvil apparatus (CS-IB, Guilin Metallurgical Machinery Co., Ltd., Guilin, China) was used in the experiment. The Bi phase transition (I-II phase transition at 2.55 GPA) was used as the fixed point to determine the pressure generated, and inserting the hot junction of a thermocouple (R-type) in the center of a heater was used to calibrate the temperature. In order to detect the phase transition of Bi, the resistance of Bi was measured by the two-wire method at room temperature. When the oil pressure is 76.5 kg/cm$^2$, the pressure reaches 2.55 GPa. Thus, the approximate calibration for reasonable pressure value at room temperature is obtained by a linear extrapolation method. The pressures were set as 1 GPa and 3 GPa. During the experiment, firstly, the graphite heater was rolled from a thin flake into a cylinder, and a gap (~2 mm) was left on it. Then, the sample blocks were assembled in the order shown in Figure 1. After, the assembly block was placed into the chamber of the six-anvil apparatus, and the pressure and the temperature were slowly increased up to the target values. The maximum temperature was guaranteed to be the same during the thermal stabilization stage so that the temperature gradient is coherent. After the samples were kept at the target pressure and temperature for 30 min, the pressure was unloaded, and the sample was taken out for analysis. The phases were characterized by a Rigaku D/max-RB X-ray diffractometer with monochromatic Cu-Kα radiation. Morphology was examined by an Olympus optical microscope (OM/6XB-PC, Shanghai, China) and a scanning electron microscope (SEM/SU8010Hitachi, Japan) operated at 15 kV equipped with an energy-dispersive X-ray spectrometer (EDS/ SU8010Hitachi, Japan).

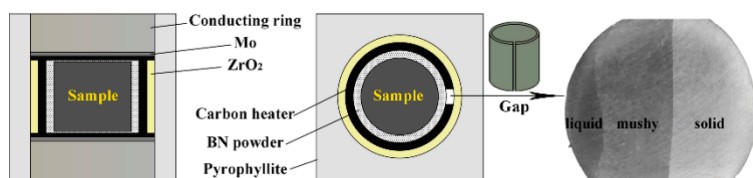

**Figure 1.** Schematic illustration of the assembly sample in the high-pressure solidification process, the structure of the graphite heater, and the final macrostructure.

## 3. Results

### 3.1. Microstructure and Phase Composition

According to the equilibrium phase diagram of Al-Ni alloy, the primary α phase is expected to precipitate first, followed by the eutectic reaction, which is demonstrated by the coupled growth

of both α-Al and β-Al₃Ni phases during the solidification of Al - 1.5 wt%Ni alloy [19]. The typical backscattered electron (BSE) microstructures of the horizontal sections of each sample solidified under different high pressures, which are displayed in Figure 2. It is worth noting that three distinct regions exist simultaneously in each sample along the directional of the temperature gradient: developed columnar dendrites in the left regions as shown in Figure 2a,d; grey solid solution, black irregular intragranular blocky microstructure, and thinner intergranular layer channels in the intermediate zones as can be seen in Figure 2c1–c3,f1–f3; and cellular dendrites in the right areas as is clear from Figure 2b,e.

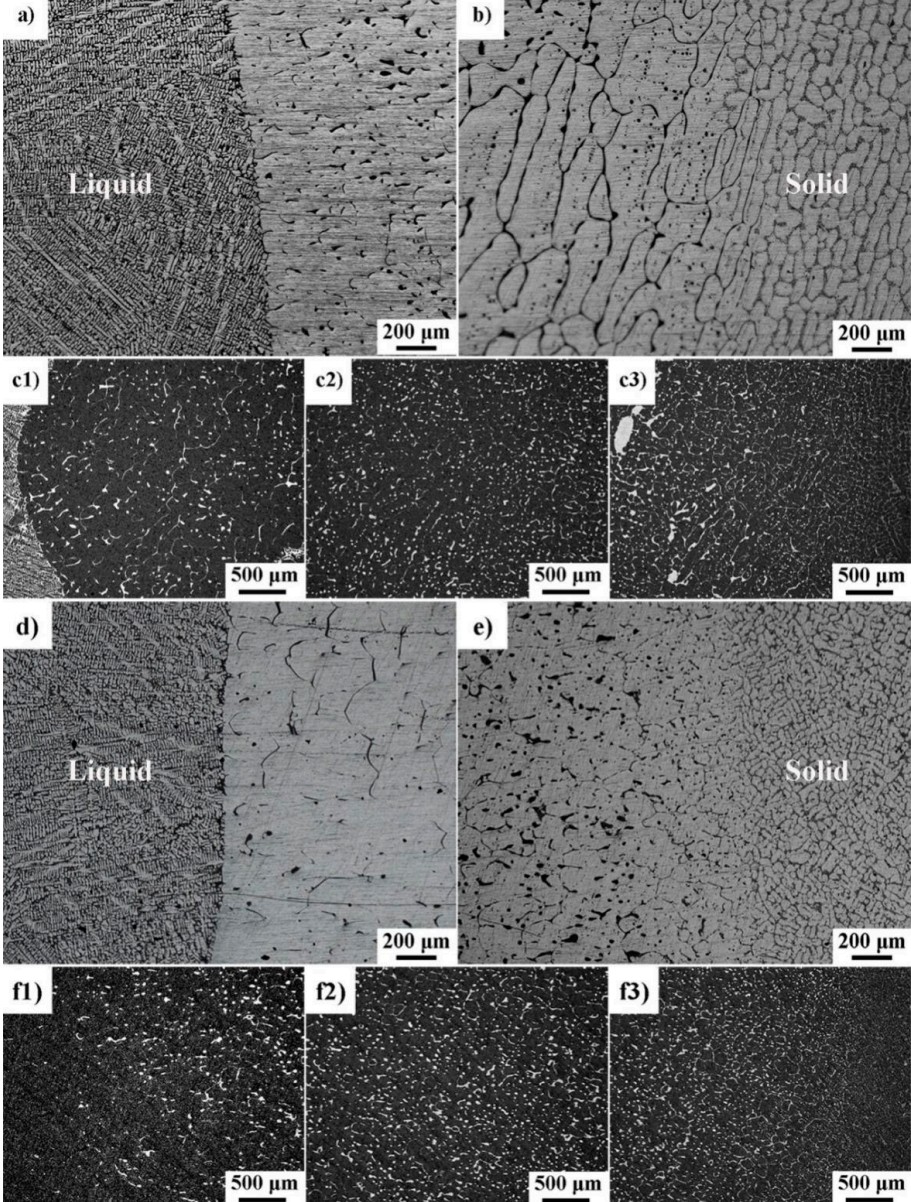

**Figure 2.** The microstructure of Al - 1.5 wt%Ni alloy solidified under different high pressures. (**a**–**c3**) 1 GPa; (**d**–**f3**) 3 GPa.

The conventional directional solidification of hypoeutectic Al - 1.5 wt%Ni alloy has been studied by Nguyen Thi [9] et al. In their study, the microstructure evolution of the partially melted region was investigated. The transverse section showed that Al grains were bound by thin eutectic layers and Ni-rich intragranular droplets. The Ni-rich droplets and the liquid channels disappeared gradually

after thermal-stable treatment of 2 h 35 min. EDS analysis of the percentage of Ni element along the growth direction showed that after the stationary growth conditions were reached, there was a sharp drop from the nominal concentration $C_0$ to nearly pure aluminum at the boundary between the non-melted zone and the partially melted region. The content of Ni element in the partially melted region increased first then decreased, whereas the content of Ni element in the fully melted region increased dramatically, and then a plateau with concentration around $C_0$ was obtained.

Therefore, with the analysis of microstructure along the temperature direction under high pressure, it is reasonable to make the conclusion that changing the structure of the graphite heater is an effective way to obtain a mushy zone under high pressure, and, here, the three regions in Figure 2 are defined as fully melted (liquid) area, mushy zone, and non-melted (solid) region.

The microstructure of Al - 1.5 wt%Ni alloy solidified at the ambient pressure is shown in Figure 3a. It shows that the eutectic microstructure presented at the grain boundaries in the form of a continuous mesh structure. The eutectic lamellar spacing was very small, and the average dendrite spacing was about 35 µm. The phase composition of Al - 1.5 wt%Ni alloys solidified under different pressures can be seen in Figure 2b. It demonstrates that the phase composition of all the alloys are the same and are composed of $\alpha$-Al and $\beta$-Al$_3$Ni phases.

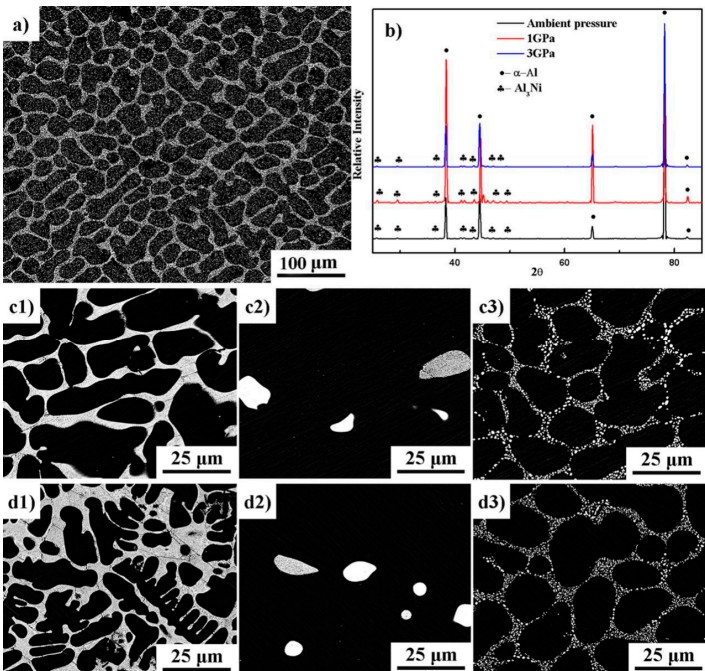

**Figure 3.** The microstructure and phase composition of Al - 1.5 wt%Ni alloy solidified under different pressures: (**a**) ambient pressure; (**b**) phase composition; (**c1–c3**) 1 GPa; (**d1–d3**) 3 GPa; (**c1,d1**) fully melted area; (**c2,d2**) mushy zone; (**c3,d3**) non-melted region.

Figure 3c1–d3 exhibits the BSE microstructures of hypoeutectic Al - 1.5 wt%Ni alloy solidified with temperature gradients and different high pressures. Figure 3c1,d1 shows the fully melted areas. As can be clearly seen, the values of secondary dendrite arm spacing under 1 GPa and 3 GPa were about 13.42 µm and 11.94 µm, respectively. Their values are smaller than that of solidification under ambient pressure. Figure 3c2,d2 displays the microstructure of mushy zones; it is clearly shown that microstructures with different brightness co-exist, and more details will be discussed later. Figure 3c3,d3 shows the non-melted regions, where the cellular $\alpha$-Al dendrite remained the same, but the eutectic lamellar spacing was obviously larger than that of solidification under ambient pressure. However, the average dendrite spacing was around 25 µm, which is smaller than the data obtained under ambient pressure.

### 3.2. The Microstructure Evolution of Mushy Zone during High-Pressure Solidification

More microstructure details of the mushy zone obtained under different pressures are explored, as displayed in Figure 4. The results reveal that four kinds of microstructure existed simultaneously (Figure 4a,b). The EDS analysis of the Ni content of the three different brightness phases in the mushy zone is described in Table 1. It demonstrates that No. 3 was the $\beta$-$Al_3Ni$ phase, No. 1 and 2 were the eutectic structures, while the Ni contents were totally different. Therefore, it is reasonable to infer that the microstructure consisted of an $\alpha$-Al phase (black matrix phase), eutectic phase (low brightness and high brightness phase), and a bright bulk $\beta$-$Al_3Ni$ phase.

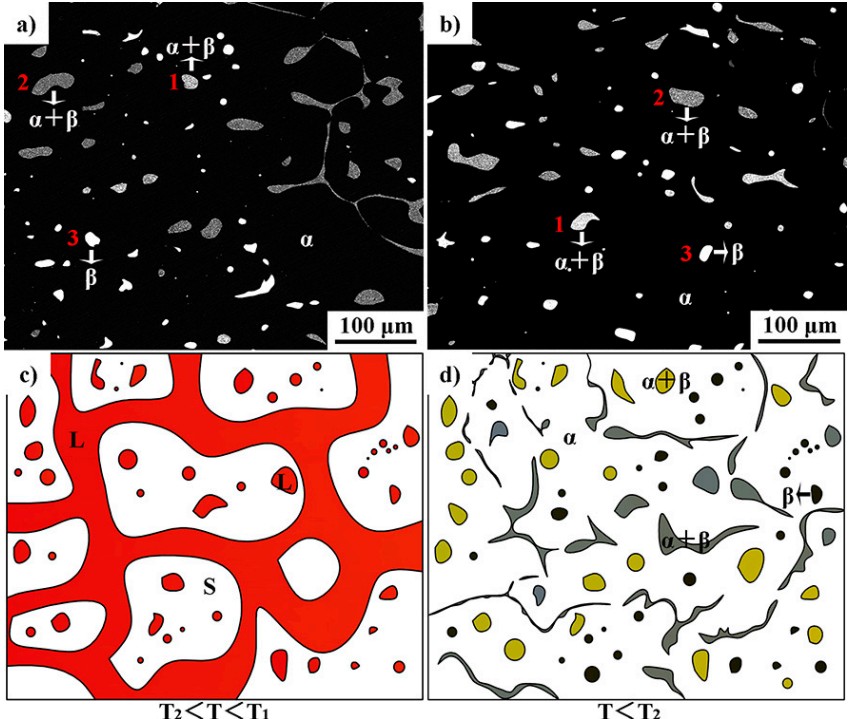

**Figure 4.** Microstructures of the mushy zone under different high pressures. (**a**) 1 GPa; (**b**) 3 GPa; (**c,d**) schematic diagram of the melting and solidification process of the mushy zone.

**Table 1.** Compositions of bright phases obtained under different pressures in the mushy zones.

| Pressure/GPa | Phase | Al/at% | Ni/at% |
|---|---|---|---|
| | 1 | 85.75 | 14.25 |
| 1 | 2 | 93.54 | 6.46 |
| | 3 | 76.19 | 23.81 |
| | 1 | 83.96 | 16.04 |
| 3 | 2 | 91.90 | 8.10 |
| | 3 | 76.17 | 23.83 |

Schematic diagrams of the melting and solidification process of the mushy zone are shown in Figure 4c,d. The reasons for these phenomena can be explained as follows. On one hand, when the temperature of the melt exceeds the eutectic temperature (Figure 4c), the eutectic phase located in the intergranular will melt and engender thinner layer channels (the transverse channel and vertical channel). Meanwhile, a thermodynamically unstable high-energy state area with defects in the intragranular and the atoms enrichment zone, caused by local diffusion of Ni atoms, will melt at the same time. This leads to the occurrence of micro-areas and isolated distribution of small liquid pools eventually, and the Ni composition may differ depending on the size. With the increase of

the temperature and prolongation of the thermal stabilization time, Ni atoms located in the liquid pool will diffuse mainly from low temperature to high temperature, which results in the melting and solidification of the liquid pool. On the other hand, when the solidification process begins (Figure 4d), the intergranular liquid solidified accompanied the precipitation of α-Al and eutectic phase. However, the situation of a liquid pool in the intragranular area is much more complicated due to the combined effects of the surrounding solid phase, which acts as the preferred substrate for nucleation of the α phase and the effect of high pressure on the eutectic point composition [15]. The eutectic transformation pattern could be divorced eutectic, which leaves the β-Al$_3$Ni nucleation and growth independently, and the cooperative growth, which exhibits higher brightness when the Ni composition of the liquid is higher.

### 3.3. The Mass Transfer in the Mushy Zone during Thermal Stabilization Treatment under High Pressure

For the solidification under certain temperature gradients, the migration of the solute-rich liquid pools and channels in the mushy zone during thermal stabilization is critical to determine the initial interface morphology and, accordingly, affects the microstructure of the fully melted area. Therefore, the understanding of the mean velocity of Ni atom migration under high pressure is a necessity. Based on the solute balance law, the velocity of atom migration can be described as follows [9]:

$$\begin{aligned} V_{mig} &= \frac{GD_L}{(k-1)m_L c_L} \\ l_{mig} &= t \cdot V_{mig} \end{aligned} \tag{1}$$

where $D_L$ is the liquid solute diffusion coefficient, $m_L$ is the liquidus slope, $c_L$ is the composition of liquid pool in the mushy zone, $k$ is the distribution coefficient, $G$ is the thermal gradient in the mushy zone, and $l_{mig}$ is the distance of the migration. In this study, $G = 30$ K/cm, $m_L = -3.52$ K/wt%, $C_0$ (1.5 wt%) $\leq C_L \leq C_E$ (5.69 wt%), and $k = 3.7 \times 10^{-3}$, $D_L = 3 \times 10^{-5}$ cm$^2$/s [23].

As is well known, high pressure exhibits significant effects on the solute diffusion coefficient; it can be expressed as below [16]:

$$D_P \approx D_L exp^{(-PV_0/RT)} \tag{2}$$

where $D_P$ is the solute diffusion coefficient under pressure (m$^2$/s), and $V_0$ is the initial molar volume of liquid (m$^3$/mol). The calculated result is shown in Figure 5a. It displays that the diffusion coefficient decreases exponentially with the increase of pressure. Take the typical values into consideration; the influences of high pressure on the migration velocity of the liquid pool are displayed in Figure 5b. So that within 30 min, 396 μm $\leq l_{mig} \leq$ 456 μm under 1 GPa, while 9 μm $\leq l_{mig} \leq$ 216 μm for the alloy solidified under 3 GPa.

Furthermore, the time required for a liquid droplet to go through the whole mushy zone can be estimated by [9]

$$\tau = \frac{T_L - T_E}{GV_{mig}} \tag{3}$$

where $T_L$ is the melting temperature (K), and $T_E$ is the eutectic temperature (K).

Substituting the physical properties of the hypoeutectic Al - 1.5%Ni alloy into Equation (3), it can be obtained that the required minimum times at 1 GPa and 3 GPa were about 746 h and 5523 h, respectively, as can be seen in Figure 5c. The situation is complex to some extent, as the application of high pressure not only effects the coefficient of solute diffusion, but it also varies the eutectic point composition to a higher value (higher melting temperature [15]). Hence, within 30 min of thermal stabilization, a large amount of the liquid pools, and channels in the mushy zone are not able to migrate to the fully melted area; moreover, the Ni-rich liquid pool in the mushy zone, which is far away from the interface, cannot influence the percentage of Ni element of the fully melted area.

The relationship between the distribution of Ni atoms in α-Al phase measured by EDS and the temperature gradient of the alloy solidified under 1 GPa is shown in Figure 5d. It demonstrates that the average Ni content of the α-Al phase in the liquid region was about 0.41 wt%, while the value in

the mushy zone and solid region decreased to 0.22 wt% and 0.26 wt%, respectively. The distribution of the Ni atoms in the mushy zone showed no downtrends. These results are different from that acquired in traditional directional solidification [9,24–26]. As is known, the distribution of Ni atoms mainly relies on the temperature gradient, while high pressure shows negative effects. Thus, its effect on the composition of the liquid region can be neglected.

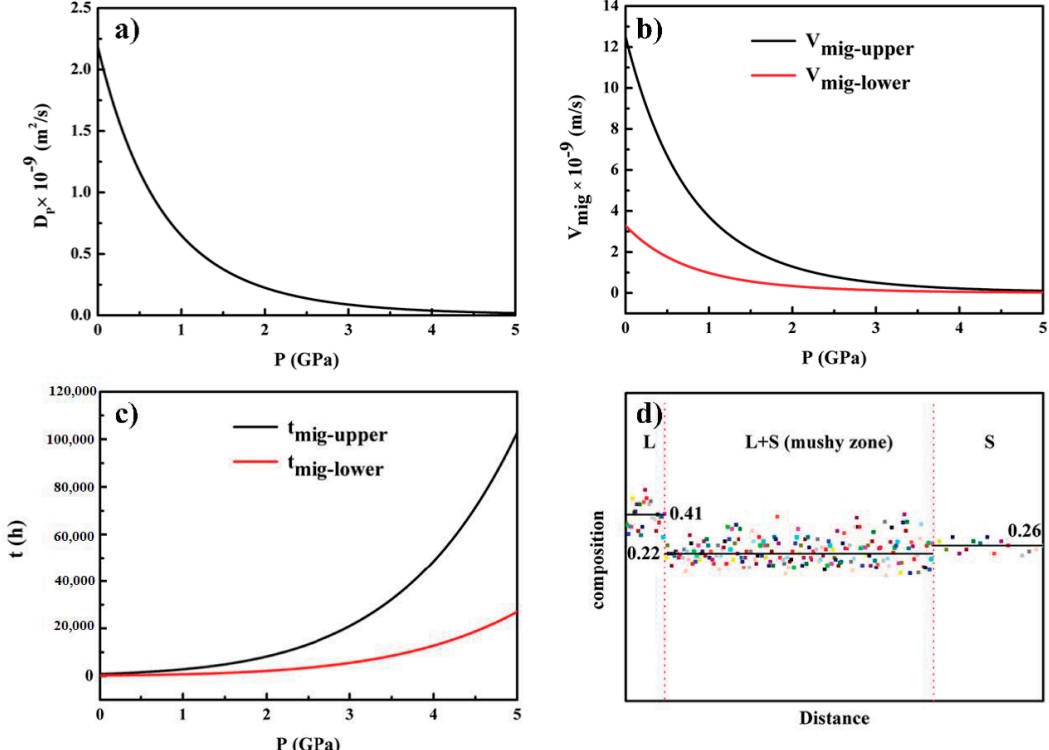

**Figure 5.** The effects of high pressure on (**a**) the coefficient of solute diffusion; (**b**) migration velocity of the liquid pool; (**c**) migration time of the liquid pool; (**d**) the content of Ni in the mushy zone.

Figure 6 is a schematic diagram of Ni atoms diffusion in the mushy zone under different conditions. It shows that the diffusion of Ni atoms in the liquid phase was greatly inhibited under pressure. Therefore, the diffusion distance of Ni atoms can be neglected in a short time. However, Ni atoms will diffuse from low temperature to high temperature at ambient pressure. If the time is long enough, a complete solid solution will be formed in the mushy zone.

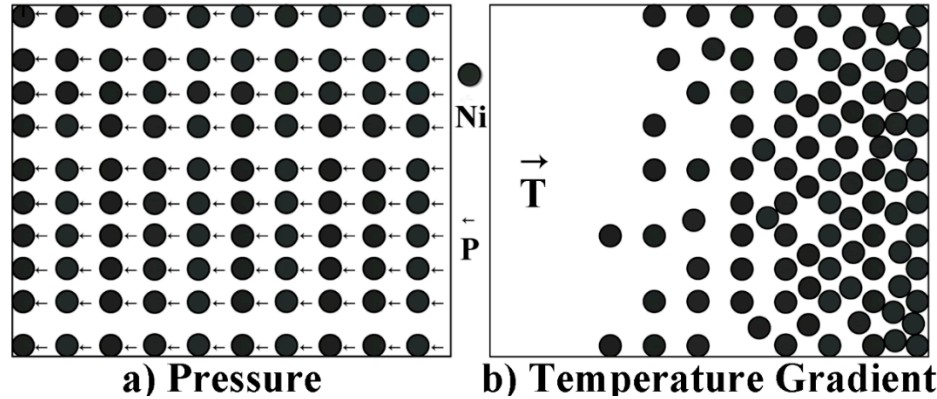

**Figure 6.** Schematic diagram of the Ni atoms diffusion in the mushy zone. (**a**) high pressure; (**b**) ambient pressure.



## 4. Conclusions

High-pressure solidification of hypoeutectic Al - 1.5 wt%Ni alloy was redesigned, and the mushy zone was obtained. In addition, the microstructure evolution and mass transfer during thermal stable treatment under different pressures were investigated.

(1) The change of the graphite heater structure is an effective way to fulfill high-pressure mushy-zone solidification.

(2) Non-melted $\alpha$-Al phase grains, intragranular Ni-rich liquid droplets, and intergranular liquid channels coexist in the mushy zone during thermal stabilization treatment. The black $\alpha$-Al phase, bright bulk Al$_3$Ni phase, and eutectic structures with different Ni content exist after solidification. Liquid channels disappeared gradually along with the temperature gradient.

(3) Mass transfer was greatly inhibited under high pressure. After thermal stabilization treatment (within 30 min), the migration distance of the liquid pool in the mushy zone at 1 GPa was between 396 μm and 456 μm, while the value was between 9 μm and 216 μm when the pressure was 3 GPa. Furthermore, the minimum time required for a liquid pool to go through the whole mushy zone at 1 GPa and 3 GPa was about 746 h and 5523 h, respectively.

**Author Contributions:** Conceptualization, X.W.; methodology, X.H.W. and D.Z.; formal analysis, D.Z.; investigation, X.W. and D.D.; resources, D.Z., H.W. and Z.W.; data curation, X.H.W.; writing—original draft preparation, X.W.; writing—review and editing, D.Z. and Z.W. All authors have read and agreed to the published version of the manuscript.

**Funding:** This work was supported by the National Natural Science Foundation of China (Nos. 51774105, 51801112, U1537201, 51501100) and the Zhejiang Province Natural Science Foundation of China (Grant No.: LY18E010003).

**Conflicts of Interest:** The authors declare no conflict of interest.

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
