# Peer review of "The Microstructure Evolution and Mass Transfer in Mushy Zone during High-Pressure Solidifying Hypoeutectic Al-Ni Alloy"

_applsci, doi:10.3390/app10207206_

Round 1
Reviewer 1 Report
Manuscript ID: applsci-944998
Title: The microstructure evolution and mass transfer in mushy zone during high pressure solidifying hypo-eutectic Al-Ni alloy
Authors: X. H. Wang et al.
1. Introduction. This section consists of 10 sentences in total. Why was this particular object (Al-Ni alloy) chosen for research? Why was the high-pressure method was used? Authors were referring to works 1988 and 2006. After 2006, there were no publications on this topic? Almost all of the sources in the Introduction are very old. Why isn't new literature being used? It is necessary to significantly expand the introduction and add information about new research on this topic. More specifically to explain the choice of alloy and research directions.
2. Materials and Methods. This section consists of one paragraph. How exactly was the sample (Al-1.5 wt% Ni alloy) prepared? Which equipment was used by the application of the high-pressure? At what temperatures? it is necessary to add the name of the companies and the country for the equipment. Authors need to separate numbers and measurement systems 20mm - 20 mm, 2mm - 2 mm, etc. Why are these high-pressure values taken?
line 61-63. Authors must add a reference for this information.
Figure 3b. What kind of peak in the diffractogram appears at the pressure of 1 GPa at 45 degrees to the right of the α-Al peak and then disappears at a pressure of 3 GPa. Why are the Al3Ni peaks at 3 GPa less intense than at 1 GPa?
lines 99-102. Size statistics are needed to confirm this information. Figure 3 does not show how these quantities were measured. Where did the numbers 13.42 microns and 11.94 microns come from? The difference is not significant. This could be a measurement error.
lines 151-152. How were these values determined?
line 165. The temperature in Kelvin or degrees? Where the temperatures came from in the article, no temperatures are indicated in the method.
Figure 5. How were these graphs built? Where did Authors get the data for pressure 2, 4 and 5 GPa, if the article only contains 1 and 3 GPa?
Technical errors:
1) For the text clarity would you refrain from using additional words, mostly meaningless words, which can be omitted or some archaic words see e.g. "respectively", "thus", "hence", "therefore", "on the other hand", "furthermore", "basically", "meanwhile", "wherein", "herein", "nonetheless", etc.
2) The bibliography does not match the style of the Applied Sciences. The names of the authors and title are written in capital letters, there is no highlighting of the names of the journals, etc. Use the Mendeley software to fix the references.
3) Don't use italic format for Figure 3 title.
Reviewer 2 Report
The study investigated the Al-1.5wt% Ni alloy. The microstructure and mass transfer in mushy zone were within the scope of the Authors' research. Organization and completeness of the work should be improved before publication. So I suggest a major revision of the work. Detailed comments are as follows.
Line 39-42: It is necessary to clarify the purposefulness of the conducted research.
Line 44: Please indicate the purity of the aluminum
Line 52-54: Please indicate the supplier of all devices in a form consistent with the requirements of the journal
Line 61: I suggest in Introduction to present a fragment of the phase equilibrium diagram of the Al-Ni system from the Al side
Figure 2:
a) The notation in the figure "liquid" and "solid" are confusing. Please change the description of "liquid side" and "solid side".
b) I also suggest dividing figure 2 into two separate ones: for 1 GPa and separate for 3 GPa. The current form is hardly to understood.
c) Please show a macroscopic image with the indication of individual zones or a diagram showing the location of individual microstructure zones. In the description of the figure, please indicate which area the observation concerns (similarly to Figure 3).
d) In the description of the figure should indicate the technique of microscopic observation (SEM, LM?)
Figure 3: The description to the figure requires re-editing. The observation technique (SEM) must be indicated in the figure description. Figure b should be included separately with the description "Results of X-ray diffraction phase identification for alloys solidified under different pressures".
Line 103 and 107: What are the secondary dendrite arm spacing and average dendrite spacing values for the solidified under ambient pressure alloy? I suggest collecting the data in the table.
Line 115: I suggest writing "eutectic structures" in place of "eutectic phases"
Line 118-135: The phenomenon is known as "constitutional liquation". A discussion on this is required.
Line 166, 197: I suggest "hypoeutectic".
Line 167: This is an estimate value, so I suggest rounding to an integer.
Line 204: Subscript is required.
Literature citations require standardization and documentation in accordance with the requirements of the journal.
Round 2
Reviewer 1 Report
The Authors made a number of corrections, but some issues were generally ignored. I'll repeat my question:
1) Materials and Methods. This section consists of one paragraph. How exactly was the sample (Al-1.5 wt% Ni alloy) prepared? Which equipment was used by the application of the high-pressure? At what temperatures? it is necessary to add the name of the companies and the country for the equipment. Authors need to separate numbers and measurement systems 20mm - 20 mm, 2mm - 2 mm, etc. Why are these high-pressure values taken?
2) The temperature in Kelvin or degrees? Where the temperatures came from in the article, no temperatures are indicated in the method.
3) The bibliography does not match the style of the Applied Sciences. The names of the authors and title are written in capital letters.
Reviewer 2 Report
- The literature list still requires correction. Some journals are italicized while others are in capital letters. The bibliography was "pasted", which was created a local background.
- Line 178: I know the meaning of "hypoeutectic". My suggestion is for the notation: "hypoeutectic" instead of "hypo-eutectic".
